# Pharmacokinetics Study of Jin-Gu-Lian Prescription and Its Core Drug Pair (*Sargentodoxa cuneata* (Oliv.) Rehd. et W and *Alangium chinense* (Lour.) Harms) by UPLC-MS/MS

**DOI:** 10.3390/molecules27134025

**Published:** 2022-06-23

**Authors:** Lin Zheng, Ting Zhou, Hui Liu, Zuying Zhou, Mingyan Chi, Yueting Li, Zipeng Gong, Yong Huang

**Affiliations:** 1State Key Laboratory of Functions and Applications of Medicinal Plants, Guizhou Provincial Key Laboratory of Pharmaceutics, Guizhou Medical University, Guiyang 550004, China; zhengl2020@126.com (L.Z.); z_tillie@163.com (T.Z.); liuhui554321@126.com (H.L.); zu_ing@163.com (Z.Z.); nhwslyt@163.com (Y.L.); 2School of Pharmacy, Guizhou Medical University, Guiyang 550004, China; naoko_568@sina.com; 3Engineering Research Center for the Development and Application of Ethnic Medicine and TCM (Ministry of Education), Guizhou Medical University, Guiyang 550004, China

**Keywords:** Jin-Gu-Lian, core drug pair, pharmacokinetics, UPLC-MS/MS

## Abstract

Jin-Gu-Lian (JGL) is traditionally used by Miao for the treatment of rheumatism arthralgia. At the same time, the combination of *Sargentodoxa cuneata* (Oliv.) Rehd. et W (SC) and *Alangium chinense* (Lour.) Harms (AC), the core drug pair (CDP) in the formula of JGL, is used at high frequencies in many Miao medicine prescriptions for rheumatic diseases. However, previous research lacks the pharmacokinetic study of JGL, and study on the compatibility of its CDP with other medicinal herbs in the formula is needed. This study aims to establish a simple, rapid, and sensitive Ultra Performance Liquid Chromatography Tandem Mass Spectrometry (UPLC-MS/MS) method for the simultaneous determination of four main bioactive components of JGL in rat plasma, including Salidroside (Sal), Anabasine (Ana), Chlorogenic Acid (CA), and Protocatechuic Acid (PCA), and compare the pharmacokinetic properties of two groups of rats after being orally administrated with JGL and its CDP extracts, respectively. The results showed that area under the plasma concentration-time curve (AUC), mean retention time (MRT), and clearance rate (CL), of Sal, Ana, CA and PCA in the two groups of rats were changed in different degrees. The CDP combined with other drugs could significantly increase the absorption of Sal and Ana, prolong its retention time in vivo, and may accelerate the absorption rate of CA and PCA. This indicated that the combination of CDP and other herbs may affect the pharmacokinetics process of active components in vivo, increase the exposure and bioavailability of compounds in the JGL group, and prolong the retention time, which may be the reason why JGL has a better inhibitory effect on inflammatory cytokines, providing a viable orientation for the compatibility investigation of herb medicines.

## 1. Introduction

The Jin-Gu-Lian (JGL) formula is a classic Miao medicine formula for the treatment of rheumatism arthralgia. In recent years, the frequency of incidence of rheumatic diseases is getting higher, especially in the southwest of China [1], which includes the settlement of the Miao people with a cold and damp climate all the year round. Over time, the Miao people gradually formed a unique theoretical system of Miao medicine for the long-term struggle with the diseases. The JGL capsule, an exclusive variety of Miao medicine for the treatment of rheumatism arthralgia syndrome in China, is the commercial product of the JGL formula, and one of the characteristic varieties developed and cultivated in the southwest of China. It is included in the *Compilation of National Standard for Traditional Chinese Medicines* and the *Medicine List for National Basic Medical Insurance and Employment Injury Insurance*. It is widely used in the treatment of rheumatic diseases such as muscle soreness, joint swelling, pain, and inhibited bending and stretching caused by rheumatism, and has a marked curative effect [2].

The clinical efficacy of Miao medicine prescription depends to a large extent on the compatibility of the formula. According to the prescription theory of traditional Chinese medicine, the drug pair is the core of compatibility and the most basic form of compatibility between different medicinal herbs in a formula [3]. Although Miao medicine is a kind of ethnic medicine with local characteristics, its prescription theory is similar to that of traditional Chinese medicine. The JGL formula is composed of five herbs, including *Sargentodoxa cuneata* (Oliv.) Rehd. et W (SC), *Alangium chinense* (Lour.) Harms (AC), *Gaultheria yunnanensis* (Franch.) Rehd (GY), *Schefflera leacantha* Vig. (SL), and *Psammosilene tunicoides* Wu, W.C. et Wu, C.Y. (PT). The combination of SC and AC, which is the core drug pair (CDP) in the JGL formula, is used at high frequencies in more than 800 Miao medicine prescriptions for rheumatic diseases collected in relevant Miao medicine books, such as *Chinese Miao Medicine*, *Chinese Materia Medica of Miao Medicinal Volume*, *Medico-Miao*, and other related Miao medicine books [4]. More significantly, CDP is not a random combination of two drugs, nor a simple accumulation of the two effects, but the sublimation of the clinical experience accumulated by ancient experts and modern practitioners [5]. The drug pair is not only the most refined compound formula, but also the core of the formula. Starting from the classical prescription, it can improve the research of single medicine to a new level; in addition, it has also established the foundation of the study of prescription [6].

Our previous work, which determined the content of the JGL capsule, showed that the contents of Salidroside (Sal), Anabasine (Ana), Chlorogenic Acid (CA) and Protocatechuic Acid (PCA) were relatively high [7,8], and they had many pharmacological activities such as relieving spasm, anti-inflammation and analgesia, skeletal muscle relaxation, and so on [9,10,11,12]. As the main active components of SC and AC, respectively, Sal and Ana are often used as exclusive components in the study of the CDP (SC and AC) [13,14]. Therefore, in this study, the four components were used as experimental indicators to study JGL and its CDP, and Figure 1 shows the chemical structures of the four compoents. Considerable work had been done on pharmacodynamics and clinical efficacy of JGL in the previous studies [15,16,17,18]. However, the research on pharmacokinetics of JGL is rare, and there is no correlation research on the prescription rule of JGL and its CDP. It is noteworthy that the crosstalk of components in JGL may affect each other’s absorption, distribution, and metabolism, incurring uncertainty regarding the efficacy and safety of this formula [19]. Therefore, in this study, the CDP of “SC + AC” was taken as the object to investigate the pharmacokinetic process of Sal, Ana, CA and PCA in vivo. On this basis, this study researched the changes of the pharmacokinetic process of the CDP after being combined with other medicinal herbs, to reveal the contribution of the drug pair in the whole prescription and the effect of compatibility on the pharmacokinetics of the four representative components in JGL. The application of SC and AC is quite representative in the compatibility of Miao medicine, and the study of its compatibility is of great significance for improving clinical efficacy, reducing toxicity and side effects, and thus extending the applied range of Miao medicine.

The purpose of this study is to explore the characteristics of compatible use of every medicinal herb in ethnic medicine formulas from the perspective of pharmacokinetics. It not only provides a reliable scientific basis for elucidating the mechanism of action and material basis of the compounds in the whole JGL formula, promoting its rational clinical application, but also explores ideas regarding the study of the pharmacokinetics of traditional Chinese medicine and its compound compatibility.

## 2. Results

### 2.1. Method Validation

#### 2.1.1. Specificity

Figure 2 shows the representative multiple reaction monitoring (MRM) chromatograms of blank plasma (A), plasma spiked with the four analytes and internal standard (IS, Levofloxacin) (B), and the samples collected from two groups of rats (C–D), including the JGL group (which was orally given JGL) and the CDP group (which was orally given the mixed extract of SC and AC). The retention time of Sal, Ana, CA, PCA and IS were 1.66, 0.63, 1.78, 1.40 and 1.99 min, respectively. All peaks obtained using the UPLC method were well separated with no interfering peaks at their respective retention times. Figure 3 shows the MS/MS spectra of the four representative analytes of JGL.

#### 2.1.2. Linearity and Lower Limits of Quantification

To assess linearity, each calibration curve was constructed with different concentrations by plotting the peak area ratio of levofloxacin versus the concentration of levofloxacin using linear regression. The typical calibration curves and linearity ranges of Sal, Ana, CA and PCA are listed in Table 1. The coefficients of correlation (*R*^2^) of all the constituents were >0.997. The lower limit of quantification (LLOQ) of the four analytes in the plasma is presented in Table 1.

#### 2.1.3. Precision and Accuracy

Precision and accuracy were determined by analyzing QC samples at three level concentrations (*n* = 5), as given in Table 2. The results showed that the intra-day and inter-day precision for the four analytes was below 15.0%, and the accuracy expressed ranged from 85.61% to 107.06% in all QC levels. All the results met the acceptance criteria. This data indicated that this developed UPLC-MS/MS method was accurate and reliable for determining compounds.

#### 2.1.4. Extraction Recovery and Matrix Effect

The four compounds’ recoveries were found to be in the range of 85.34% to 106.48% with RSD values less than 12.47%, indicating that the recoveries of the analytes were consistent and reproducible. Moreover, the results of the matrix effects suggested no significant ion suppression or enhancement in this UPLC method, as shown in Table 3.

#### 2.1.5. Stability

The analyte stability in plasma was demonstrated by analyzing processed plasma samples of QC samples at three level concentrations (*n* = 6) at three different storage conditions (Table 4). It showed that Sal, Ana, CA, and PCA were stable but under restricted storage conditions due to their sensitive nature. The precision was found for room temperature from 2.06% to 8.44% with accuracy (89.91~108.63%), for refrigeration stability conditions from approximately 4.77% to 12.19% with accuracy (91.90% to 108.53%), and for freeze-thaw stability from 3.81% to 10.62% with accuracy (91.75~107.43%).

### 2.2. Pharmacokinetic Study

The validated UPLC-MS/MS method was applied to investigate the pharmacokinetic profiles for simultaneous determination of the four analytes (Sal, Ana, CA and PCA) in two groups of rats, including the JGL group (which was orally given JGL extract) and the CDP group (which was orally given the mixed extract of SC and AC). The mean plasma concentration–time profiles of Sal, Ana, CA and PCA are displayed in Figure 4. In addition, the main plasma pharmacokinetic parameters of the two groups of rats are summarized in Table 5.

An independent sample t-test was applied to compare the differences of pharmacokinetic parameters of Sal, Ana, CA, and PCA, including the time to reach the maximum concentration (Tmax), maximum plasma concentration (Cmax), and area under concentration-time curve (AUC), clearance rate (CL), apparent volume of distribution (V), and mean retention time (MRT). In comparison with the CDP group, the AUC_(0-t)_ and MRT_(0-t)_ of Sal in the JGL group were significantly higher than those in the CDP group (*p* < 0.05, *p* < 0.01, respectively), and the MRT_(0-t)_ of Ana in the JGL group were significantly higher than those in the CDP group (*p* < 0.05), which indicated that the CDP combined with other drugs could significantly increase the absorption of Sal and Ana, and prolong its retention time in vivo. Although Ana is the main active ingredient of SC in CDP, it demonstrates distinct effects on relaxing muscle and soothing for pain, but at the same time, it also has certain toxicity [20]. The C_max_ and V_z/F_ of Ana was significantly lower in the JGL group (*p* < 0.05, *p* < 0.01, respectively); Ana is a kind of alkaloid with good lipophilicity, which easily crosses the biological membrane and distributes to various tissues. The decrease of C_max_ and V_z/F_ in the JGL group may help to avoid the accumulation of Ana in a certain tissue, which may help to reduce the possibility of toxicity of Ana. It is helpful for the treatment of SC when CDP is combined with other medicinal materials. CA and PCA, which are not proprietary ingredients in the CDP, also experienced certain changes in several pharmacokinetic parameters. These changes may play a certain role in the efficacy of the JGL formula. Compared with the CDP group, the C_max_ of CA increased significantly (*p* < 0.05), and V_z/F_ of CA had a highly significant rise in the JGL group (*p* < 0.01), but there was no significant difference in the AUC and MRT between the two groups. Meanwhile, the T_max_ of PCA was significantly later (*p* < 0.05), and AUC increased with no significant difference in the JGL group. This suggested that the combination of various herbs may accelerate the absorption rate of CA and PCA, but had little effect on its extent of absorption.

As Sal and Ana are considered the dominant effective components in SC and AC due to relatively higher content and bioactivity [21,22], the exposure of Sal in vivo was the highest and the clearance rate was the lowest among the four representative components, and the exposure of Ana was also relatively high whether in the JGL group or the CDP group. This suggested that the CDP may be absorbed best, compared with other medicinal herbs in JGL. According to the concentration–time curve, Sal, the specific component of SC, showed a bimodal effect in the JGL group, but there was only a single peak in the CDP group, and there was no bimodal effect in the previous pharmacokinetic studies of Sal [23,24]. The bimodal effect may be caused by drug–drug interaction when SC is used in combination with AC, GY, SL and PR. In addition, Sal has active efflux in the process of absorption [25], so the bimodal phenomenon could be caused by the inhibition of Sal efflux enzyme after the compatibility of other drugs in JGL.

Except for Ana, the values of T_max_ of Sal, CA and PCA were within 0.92 h after oral administration. This showed that these three analytes were absorbed quickly, while Ana experienced less absorption. Teng-Fei Chen et al. have studied the pharmacokinetics of salidroside in vivo after being orally administrated with the extract of the JGL capsule by HPLC-MS/MS [26]. The results showed that salidroside reached the highest concentration after 0.80 h, and the MRT_(0-t)_ was 1.75 ± 0.07 h, which is consistent with the results of this study. This suggested that the dynamic process of JGL in vivo is stable and can be recreated. However, their research only determined Sal in JGL, which cannot fully reflect the in vivo process of all herbs in JGL. On the other hand, the purpose of their research is different from this study, so the concept of CDP is not involved in their study. In this study, a novel UHPLC-MS/MS-based bioanalytical method was developed successfully, with the least range and also high sensitivity and efficiency. This method also showed various advantages in terms of being very economical, with a less than 5 min retention time, and successfully validated. This method was used successfully to evaluate the pharmacokinetics of JGL and its CDP (SC and AC).

To explore the role of the CDP (SC and AC) in the whole prescription in our previous basic research on the JGL formula, related inflammatory factors (RF, TNF-α, IL-6, IL-1β, PEG2 and 5-HT) were determined in rat plasma of JGL group and CDP group, respectively. The results showed that both the two groups had a certain inhibitory effect, and that this inhibitory effect of the JGL group was much better; this indicated that the CDP had a better effect on compatibility with other medicinal materials in the JGL formula. Generally speaking, comparing with the CDP group, the pharmacokinetic characteristics of the JGL group were characterized by increased exposure and bioavailability, which prolonged the residence time in vivo. This suggested that the pharmacokinetic process of its representative components in vivo was affected after being combined with GY, PR and SL, which may be the reason why JGL has a better inhibitory effect on inflammatory cytokines when the CDP is combined with the other herbs in the whole compound prescription, and provided the experimental data and theoretical basis for further development and clinical applications of the CDP.

## 3. Materials and Methods

### 3.1. Materials and Reagents

Reference substances salidroside (product no. 181020) and DL-anabasine (product no. 21J064-C2) were obtained from the Beijing Century Aoke Biotechnology Co., Ltd. (Beijing, China). Chlorogenic acid (product no. 110753–201716) was purchased from the China National Institutes for Food and Drug Control (Beijing, China). Protocatechuic acid (product no. MUST-20110310) was procured from Chengdu Must Bio-Technology Co., Ltd. (Chengdu, China). JGL extract (product no. 190806) and medicinal materials of the CDP group were procured from Guizhou Yibai Pharmaceutical co., LTD (Guiyang, China). Heparin sodium (product no. 125P028) was procured from Beijing Solarbio Science & Technology co., Ltd. (Beijing, China). High-performance liquid chromatography grade acetonitrile and methanol were purchased from Merck & Co., Inc. (Darmstadt, Germany). Formic acid was purchased from Thermo Fisher Scientific Inc. (Shanghai, China). Distilled water was obtained from Guangzhou Watson’s Food & Beverage Co., Ltd. (Guangzhou, China). All other solvents used were of analytical grade and are commercially available.

### 3.2. Animals

Healthy male Sprague-Dawiey rats (220 ± 20 g) were provided by Changsha Tianqin Biotechnology Co., Ltd. (Beijing, China) (license number: SCXK (Xiang) 2019–0014). Prior to the experiments, the rats were acclimatized for a week in an animal room under air conditioning (22 ± 2 °C) and an auto-controlled photoperiod of 12:12 h light/dark cycle. The rats were free to access the water and food until 12 h before the experiment with constant temperature (20–25 °C), humidity (50%).

### 3.3. Assay of JGL Extract and the Mixed Extract of SC and AC

The composition and preparation method of the Jin-Gu-Lian formula were as follows [27]: 50 g of *Psammosilene tunicoides* W.C. Wu et C.Y. Wu (PT), 343 g of *Gaultheria yunnanensis* (Franch.) Rehd (GY), 400 g of *Schefflera leacantha* Vig. (SL), 50 g of *Alangium chinense* (Lour.) Harms (AC) and 370 g of *Sargentodoxa cuneata* (Oliv.) Rehd. et W (SC), respectively. After boiling for 3 h, the filtrate of SL was filtered, and the residue was dried in the oven and crushed into fine powder; the other four herbs were decocted with water, filtered, and combined with the filtrate of PT, concentrated to a thick extract. Finally, the fine powder of PT was added in proportion (extract: powder = 8.7:1) to get the final product. Likewise, the CDP extracts were prepared according to the same proportion and preparation method of the JGL extract. The contents of Sal, Ana, CA and PCA were determined by UPLC-MS/MS). In the JGL extract, the contents of Sal, Ana, CA and PCA were 0.0029, 0.4672, 1.1691 and 0.3112 mg/g, respectively. In the SC and AC extract, the contents of Sal, Ana, CA and PCA were 0.0106, 1.0435, 1.0816 and 0.3874 mg/g, respectively.

### 3.4. Preparation of Calibration Standards and Quality Control (QC) Solutions

The stock solutions of four analytes were prepared in methanol. The IS stock solution (1.052 mg/mL) was prepared in methanol and diluted to a final concentration of 105.2 ng/mL with methanol before analysis, and stored at 4 °C. The mixed stock solution was prepared by serial dilution of each individual stock solution with methanol. Mix calibration standard samples containing Sal from 13.10 to 1048.0 ng/mL, Ana from 0.1125 to 28.8 ng/mL, CA from 1.85 to 118.2 ng/mL, and PCA from 1.6172 to 258.750 ng/mL were obtained by spiking the appropriate working solution into blank plasma. In terms of the validation and pharmacokinetic study of the assay, three (low, medium and high) concentrations of the standard solution, including Sal (26.20, 104.80 and 419.20 ng/mL), Ana (0.23, 3.60 and 14.40 ng/mL), CA (3.69, 14.78, and 59.10 ng/mL), and PCA (3.23, 12.94 and 103.50 ng/mL) were applied to be the quality control (QC) samples. The standard solutions and QC samples were extracted on each analysis day with the same processes for plasma samples prepared as the description below.

### 3.5. Plasma Sample Preparation

To 100 µL of rat plasma sample, 50 µL of 1% formic acid was added and then vortexed for 0.5 min, then 20 µL of IS solution (105.2 ng/mL) was added and then vortexed for 0.5 min. The mixture was extracted with 400 µL of methanol and vortexed for 5 min. After ultrasound for 10 min and centrifugation at 12,000× *g* for 10 min, the supernatant was transferred into another clean Eppendorf tube quantitatively and evaporated to dryness at 37 °C with nitrogen. The residue was reconstituted in 150 µL initial mobile phase. After ultrasound for 10 min and centrifugation at 14,000× *g* and 4 °C for 10 min, the supernatant was injected into the UPLC-MS/MS system for analysis.

### 3.6. Instruments and Analytical Condition

The Acquity UPLC (TM) system tandem MS was used for separation and detection. The column used for chromatographic separation was a Waters Acquity BEH C_18_ column (2.1 × 50 mm, 1.7 µm; Waters, Wexford, Ireland). The system was also equipped with a Waters VanGuard BEH C_18_ (2.1 × 5 mm, 1.7 µm) column. The mobile phase consisted of water contained with 0.2% formic acid (A), as well as acetonitrile contained with 0.2% formic acid (B). The elution gradient was the same as the following sections (Table 6). The flow rate was 0.3 mL/min under 40 °C of the column temperature. The injection volume was 3 µL.

The Waters TQS Quantum triple quadrupole mass spectrometer equipped with electrospray ionization (ESI) source was used for mass analysis and detection. The mass spectrometer was operated in either the positive or negative mode, and the multiple reaction monitoring (MRM) mode was selected to quantify the reference substrate (Table 7). Data acquisition and processing was performed using Micromass Masslynx version 4.1, and the reference substrate was quantified according to a validated UPLC-MS/MS method. The main operating parameters were set as follows: desolvation temperature, 400 °C; nebulizer gas (N2), 800 L/h; source heater, 150 °C; capillary ionization voltage: 1.0 kV.

### 3.7. Method Validation

#### 3.7.1. Specificity

The specificity of the method was assessed by comparing chromatograms of three different plasma samples: six individual blank plasma samples (A), blank plasma spiked with Sal, Ana, CA, PCA and IS (B), and plasma samples obtained after the oral administration of JGL extract (C1) and the mixed extract of SC and AC (C2).

#### 3.7.2. Linearity and Lower Limits of Quantification

Linearity curves were obtained by assaying standard calibration samples at six concentration levels. The linearity of each calibration curve was determined by plotting the peak area ratio (Y) of four analytes to the IS versus the concentrations (X) of analytes with weighted least square linear regression. The LLOQ determined based on the analyte response should be at least 10 times that of the blank response.

#### 3.7.3. Precision and Accuracy

The intra-day and inter-day (*n* = 5) precision and accuracy were evaluated using three different QC samples for each analyte. The precision was expressed as the relative standard deviation (RSD, %) from the theoretical concentrations.

#### 3.7.4. Extraction Recovery and Matrix Effect

Recoveries of the four analytes from plasma were calculated by comparing the peak areas of pretreated QC samples with those of post-extracted blank plasma samples spiked with the analytes at the same concentration. Matrix effects were determined at three QC levels by comparing the peak areas obtained from blank blood extract spiked with the four analytes to those of pure standard solutions containing the same amount of the analytes.

#### 3.7.5. Stability

The stability test was conducted using three different concentrations of QC samples under various conditions. The short-term stability was evaluated by analyzing samples kept at 20 °C for 24 h. The freeze-thaw stability was assessed over three freeze-thaw cycles (−80 °C to 25 °C). In addition, the stability was confirmed at 8 °C for 24 h.

### 3.8. Pharmacokinetic Study

Twelve rats with similar average body weight were divided into two groups (six animals per group); the JGL group (was orally given JGL extract) and the CDP group (was orally given the mixed extract of SC and AC). The JGL extract and the mixed extract of SC and AC were, respectively, dissolved in 0.5% carboxymethyl cellulose sodium (CMC-Na) aqueous solution for oral administration; then, they were administered to rats orally with a single dose of 6.8670 g/kg and 2.1323 g/kg, respectively. For each rat, the tail venous blood samples (0.25 mL) were obtained into heparinized polythene tubes before drug administration and at different time points of 0.0833, 0.1667, 0.3333, 0.5, 0.67, 1, 1.5, 2, 4, 6, 8, 12, 24, and 36 h post-dosing. Subsequently, the blood samples were centrifuged at 6000× *g* for 6 min at 4 °C to separate the plasma. Finally, the plasma was transferred to clean Eppendorf tubes and stored at −20 °C until analysis.

### 3.9. Data Analysis

To determine the pharmacokinetic parameters of the four active ingredients, the concentration–time data were analyzed using the WinNonLin 8.2 pharmacokinetic program (Pharsight Corporation, Phoenix, AZ, USA) with the non-compartmental method. The comparisons of pharmacokinetic data between the two groups were performed using the SPSS (IBM SPSS Statistics 20.0 Developer, IBM Corp, New, York, NY, USA) software with analysis of variance. The results were compared by independent sample t-test. All the results are expressed as arithmetic mean ± standard deviation (SD).

## 4. Conclusions

In conclusion, the study established and validated a simple, rapid, and sensitive UPLC-MS/MS method for the simultaneous determination of four bioactive components in rat plasma. The approach was successfully applied to study the plasma pharmacokinetics following oral administration of JGL and its CDP extract, respectively. The study compared the pharmacokinetic differences of the representative components of JGL between the CDP and the whole prescription, which was expected to clarify the rationality of compatibility of JGL at the pharmacokinetic level for the first time. Overall, the pharmacokinetic process of its representative components in vivo was affected after being combined with GY, PR and SL. These results provide a reference for the future study of clinical compatibility and mechanism of this commonly used CDP of SC and AC. The function and mechanism of CDPs in traditional Chinese medicine formulas is currently not clear; if it is studied clearly, it will be of certain significance to inherit and develop the compatibility theory of traditional Chinese medicine. More importantly, it is helpful for guiding the clinical medication and making the research and development of new drugs of traditional Chinese medicine more effective.

## Figures and Tables

**Figure 1 molecules-27-04025-f001:**
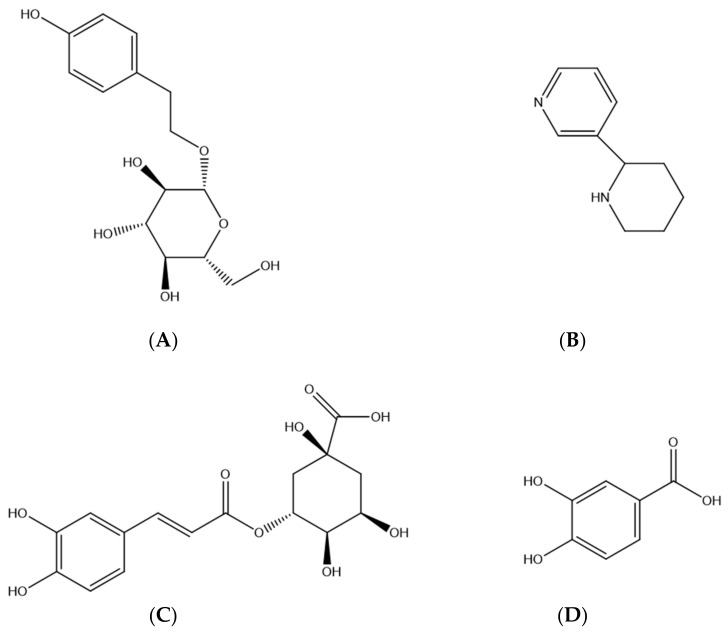
Chemical structures of Salidroside (**A**), Anabasine (**B**), Chlorogenic Acid (**C**), and Protocatechuic Acid (**D**).

**Figure 2 molecules-27-04025-f002:**
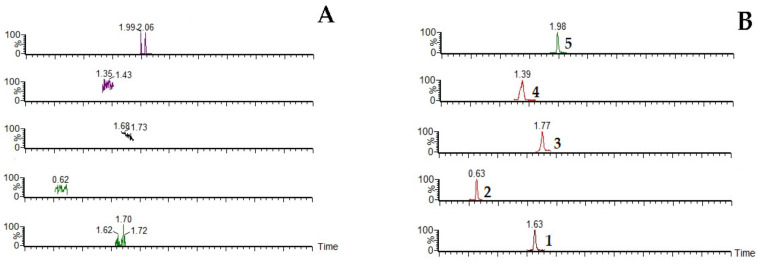
Representative multiple reaction monitoring (MRM) chromatograms of blank plasma (**A**); plasma spiked with the four analytes and IS (**B**); and the samples collected from Jin-Gu-Lian (JGL) group (**C**) and the core drug pair (CDP) group (**D**). 1. Salidroside; 2. Anabasine; 3. Chlorogenic Acid; 4. Protocatechuic Acid; 5. Levofloxacin.

**Figure 3 molecules-27-04025-f003:**
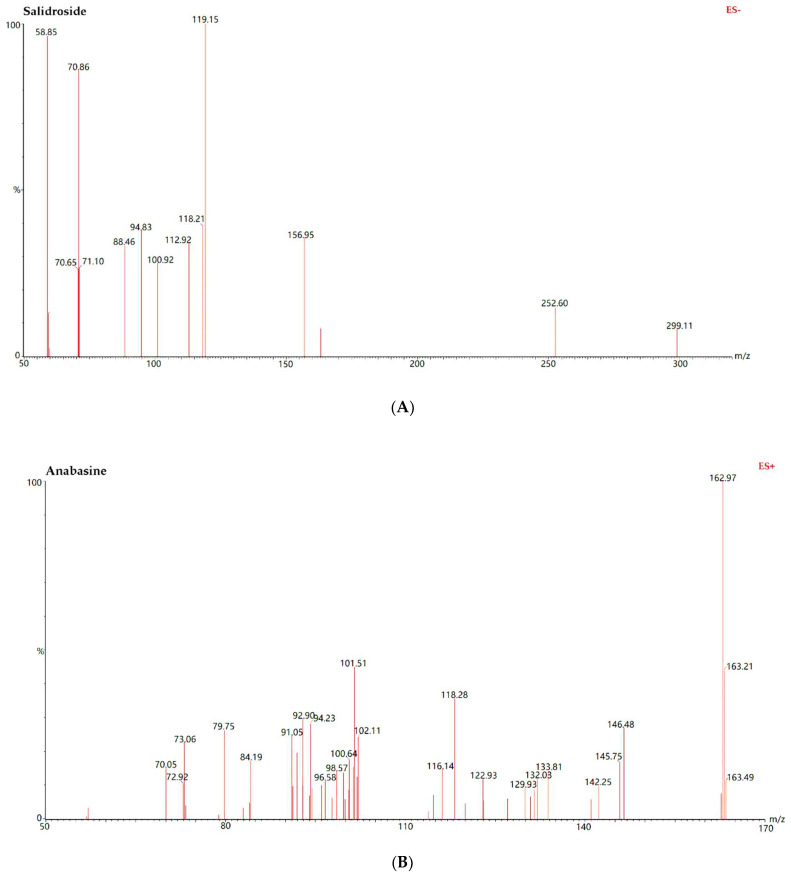
MS/MS spectra of (**A**) Salidroside; (**B**) Anabasine; (**C**) Chlorogenic Acid; (**D**) Protocatechuic Acid. ES−: Electron pray ionization with negative ion mode; ES+: electron pray ionization with postive ion mode; MS: Mass Spectra.

**Figure 4 molecules-27-04025-f004:**
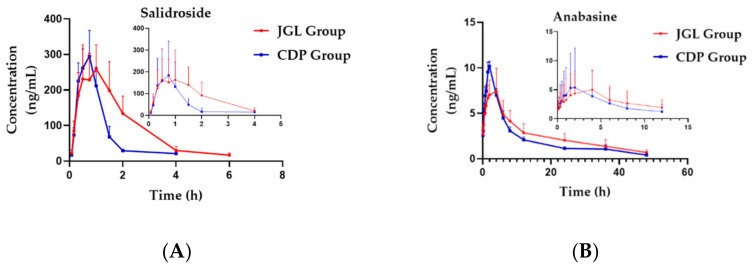
(**A**) Comparison of mean plasma concentration–time profiles of Sal in two groups; (**B**) comparison of mean plasma concentration–time profiles of Ana in two groups; (**C**) comparison of mean plasma concentration–time profiles of CA in two groups; (**D**) comparison of mean plasma concentration–time profiles of PCA in two groups (*n* = 6).

**Table 1 molecules-27-04025-t001:** Linearity data of the four analytes in rat plasma.

Analytes	Regression Equation	*R* ^2^	Calibration Range (ng/mL)	LLOQ (ng/mL)
Salidroside	*y* = 0.0007*x* + 0.0038	0.9979	13.10~1048	13.10
Anabasine	*y* = 0.0203*x* + 0.0779	0.9986	0.1125~28.80	0.1125
Chlorogenic Acid	*y* = 0.0392*x* − 0.0218	0.9976	1.850~118.2	1.850
Protocatechuic Acid	*y* = 0.0180*x* − 0.0079	0.9996	1.617~258.8	1.6172

LLOQ: Lower Limit of Quantification.

**Table 2 molecules-27-04025-t002:** Intra- and inter-day precision and accuracies of the analytes in rat plasma (*n* = 5).

Analytes	Spiked Conc. (ng/mL)	Intra-Day			Inter-Day		
		Measured Conc. ^a^ (ng/mL)	Precision	Accuracy	Measured Conc. ^a^ (ng/mL)	Precision	Accuracy
			(RSD)%	(%)		(RSD)%	(%)
Salidroside	26.20	26.84 ± 1.17	4.35	102.46	25.10 ± 1.81	7.23	95.80
	104.80	96.67 ± 8.36	8.65	92.25	108.42 ± 6.66	6.14	103.45
	419.20	384.46 ± 38.71	10.07	91.71	392.19 ± 41.64	10.62	93.56
Anabasine	0.23	0.20 ± 0.01	7.21	87.87	0.21 ± 0.02	9.01	91.34
	3.60	3.32 ± 0.18	5.40	92.26	3.72 ± 0.12	3.11	103.28
	14.40	14.09 ± 1.12	7.97	97.86	15.07 ± 1.15	7.65	104.62
Chlorogenic Acid	3.69	3.30 ± 0.19	5.74	89.47	3.30 ± 0.20	6.10	89.52
	14.78	13.69 ± 0.60	4.40	92.64	14.60 ± 1.08	7.37	98.81
	59.10	61.30 ± 4.30	7.02	103.72	63.27 ± 4.14	6.54	107.06
Protocatechuic Acid	3.23	3.33 ± 0.11	3.43	103.25	3.34 ± 0.28	8.42	103.54
	12.94	12.25 ± 0.84	6.83	94.68	13.64 ± 0.83	6.12	105.44
	103.50	94.15 ± 6.01	6.38	90.96	98.26 ± 7.55	7.68	94.94

^a^ Mean ± standard deviation. RSD: relative standard deviation

**Table 3 molecules-27-04025-t003:** Extraction recoveries and matrix effects of the four analytes in rat plasma (*n* = 5).

Analytes	Spiked Conc. (ng/mL)	Recovery		Matrix Effect	
		Mean ± SD (%)	RSD%	Mean ± SD (%)	RSD%
Salidroside	26.20	85.34 ± 5.66	6.63	94.14 ± 7.04	7.48
	104.80	95.10 ± 8.60	10.70	106.65 ± 7.42	6.96
	419.20	104.01 ± 7.41	7.12	103.96 ± 5.01	4.82
Anabasine	0.23	91.64 ± 4.35	4.74	88.52 ± 6.75	7.63
	3.60	94.68 ± 5.31	5.61	102.45 ± 2.57	2.51
	14.40	95.98 ± 7.70	8.02	108.13 ± 8.97	8.30
Chlorogenic Acid	3.69	86.05 ± 5.33	6.20	105.29 ± 9.25	8.79
	14.78	94.83 ± 6.79	7.16	101.86 ± 3.25	3.19
	59.10	93.94 ± 4.83	5.15	94.56 ± 6.74	7.13
Protocatechuic Acid	3.23	90.14 ± 7.87	8.73	89.93 ± 8.46	12.47
	12.94	103.10 ± 4.76	4.62	93.61 ± 6.50	6.95
	103.50	106.48 ± 5.20	4.88	107.84 ± 9.39	8.70

**Table 4 molecules-27-04025-t004:** Stability of the four analytes in rat plasma samples (*n* = 5).

Analytes	Spiked Conc. (ng/mL)	Room Temperature Stability			Refrigeration Stability			Freeze-Thaw Stability		
		Measured Conc. ^a^ (ng/mL)	RSD%	Accuracy (%)	Measured Conc. ^a^ (ng/mL)	RSD%	Accuracy (%)	Measured Conc. ^a^ (ng/mL)	RSD%	Accuracy (%)
Salidroside	26.20	28.46 ± 1.29	4.54	108.63	27.71 ± 1.32	4.77	105.75	26.77 ± 1.88	7.02	102.17
	104.80	112.08 ± 6.28	5.60	106.94	101.93 ± 8.62	8.46	97.26	110.4 ± 7.36	6.66	105.35
	419.20	405.56 ± 27.81	6.86	96.75	400.97 ± 22.00	5.49	95.65	402.81 ± 18.34	4.55	96.09
Anabasine	0.23	0.23 ± 0.01	4.74	103.14	0.24 ± 0.03	12.19	108.53	0.21 ± 0.02	10.62	94.57
	3.60	3.44 ± 0.28	8.23	95.43	3.48 ± 0.30	8.49	96.57	3.87 ± 0.15	3.81	107.43
	14.40	12.95 ± 0.85	6.53	89.91	13.67 ± 0.79	5.74	94.93	15.05 ± 0.98	6.53	104.52
Chlorogenic Acid	3.69	3.82 ± 0.25	6.50	103.43	3.93 ± 0.22	5.64	106.41	3.41 ± 0.21	6.21	92.37
	14.78	13.82 ± 1.17	8.44	93.51	13.91 ± 0.75	5.42	94.10	15.06 ± 0.84	5.59	101.87
	59.10	60.75 ± 3.88	6.38	102.78	58.04 ± 5.53	9.53	98.20	61.16 ± 3.38	5.53	103.49
Protocatechuic Acid	3.23	3.37 ± 0.11	3.16	104.13	3.43 ± 0.25	7.36	106.25	3.08 ± 0.20	6.38	95.35
	12.94	12.39 ± 0.26	2.06	95.79	11.89 ± 1.07	9.02	91.90	13.43 ± 0.54	3.99	103.82
	103.50	105.03 ± 5.94	5.66	101.48	105.06 ± 12.80	12.18	101.50	94.97 ± 5.29	5.57	91.75

^a^ Mean ± standard deviation.

**Table 5 molecules-27-04025-t005:** Pharmacokinetic parameters of rats in the JGL group and the CDP group. Each point represents the mean ± SD (*n* = 6).

Parameters	Group	Sal	Ana	CA	PCA
T_max_ (h)	JGL	0.72 ± 0.31	2.83 ± 1.33	0.49 ± 0.15	0.92 ± 0.38 *
	CDP	0.71 ± 0.10	1.83 ± 0.26	0.63 ± 0.21	0.49 ± 0.15
C_max_ (ng/mL)	JGL	285.08 ± 76.24	7.76 ± 2.31 *	21.93 ± 6.81 *	63.45 ± 21.21
	CDP	298.98 ± 67.05	10.42 ± 0.44	14.52 ± 3.68	87.42 ± 21.68
AUC_(0-t)_ (hr × ng/mL)	JGL	588.27 ± 180.86 *	122.57 ± 38.72	54.09 ± 17.76	210.88 ± 82.57
	CDP	347.83 ± 68.20	102.16 ± 5.53	53.24 ± 7.21	154.45 ± 33.25
AUC_(0-∞)_ (hr × ng/mL)	JGL	620.33 ± 187.46 *	140.39 ± 46.21	81.34 ± 26.26	238.54 ± 85.62
	CDP	404.08 ± 88.52	111.83 ± 11.39	72.65 ± 10.19	201.88 ± 32.51
CL_z/F_ (L/h/kg)	JGL	3.73 ± 1.20 *	0.15 ± 0.04 **	107.87 ± 36.04 **	14.89 ± 5.00
	CDP	2.13 ± 0.48	0.20 ± 0.02	32.24 ± 4.21	11.28 ± 1.98
V_z/F_ (L/kg)	JGL	6.98 ± 2.75	3.72 ± 1.33 **	1129.06 ± 433.16 **	203.82 ± 145.87
	CDP	5.01 ± 2.69	4.21 ± 1.06	318.18 ± 67.43	261.77 ± 219.30
MRT_(0-t)_ (h)	JGL	1.68 ± 0.14 **	15.15 ± 1.52 *	4.04 ± 0.37	5.42 ± 0.58
	CDP	1.10 ± 0.09	12.93 ± 1.26	4.41 ± 0.25	5.20 ± 1.14
MRT_(0-∞)_ (h)	JGL	2.01 ± 0.26	22.49 ± 4.52	10.61 ± 3.98	9.42 ± 1.61
	CDP	1.92 ± 0.84	17.86 ± 4.68	9.23 ± 1.73	16.50 ± 9.41

JGL: Jin-Gu-Lian; CDP: core drug pair; T_max_: the time to reach the maximum concentration; C_max_: maximum plasma concentration; AUC: area under the plasma concentration-time curve; CL: clearance rate; V: apparent volume of distribution; MRT: mean retention time. * *p* < 0.05, ** *p* < 0.01

**Table 6 molecules-27-04025-t006:** Elution gradient.

Time (min)	%A	%B	Curve
Initial	95	5	Initial
0.5	95	5	6
2.5	60	40	6
3.5	5	90	6
4.5	5	95	6
5	95	95	1
5.5	95	95	1

**Table 7 molecules-27-04025-t007:** List of selected MRM parameters, cone voltage (CV), collision energy (CE), and retention time of each analyte and internal standard (IS).

Analyte	Polarity	Parent Ion (Da)	Daughter Ion (Da)	CV (V)	CE (eV)
Salidroside	ESI−	299.1	119.0	35	20
Anabasine	ESI+	163.4	94.2	30	15
Chlorogenic Acid	ESI−	353.0	191.0	30	15
Protocatechuic Acid	ESI−	152.9	109.0	30	15
Levofloxacin (IS)	ESI−	362.2	261.1	20	10

## Data Availability

The data presented in this study are included within the article.

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
