# Peer review of "Pharmacokinetics Study of Jin-Gu-Lian Prescription and Its Core Drug Pair (Sargentodoxa cuneata (Oliv.) Rehd. et W and Alangium chinense (Lour.) Harms) by UPLC-MS/MS"

_molecules, 2022, doi:10.3390/molecules27134025_

Round 1

Reviewer 1 Report

The manuscript presented for review "Pharmacokinetics Study of Jin-Gu-Lian Prescription and Its Core Drug Pair (Sargentodoxa cuneata (Oliv.) Rehd. et W and Alangium chinense(Lour.)Harms) by UPLC-MS/MS" contains interesting research results of great practical importance. The aim of the research was to develop and validate a simple, rapid, sensitive UPLC-MS/MS method for the simultaneous determination of four main bioactive components of JGL in rat plasma, including Salidroside (Sal), Anabasine (Ana), 23 Chlorogenic Acid (CA), and Protocatechuic Acid (PCA), compare the pharmacokinetic properties of two groups of rats after being orally administrated JGL and its core drug pair extracts, respectively. The manuscript is well written and suitable for publication after some revisions suggested below:
Abstract: The quantitative information is completely missing. Authors are advised to include some quantitative information in order to enhance the readability of the manuscript.

Abbreviations: Authors are advised to define each abbreviation in first appearance in abstract, text, table and figure.

Introduction: Introduction is too lengthy. Please remove general information in order to shorten the length of Introduction.

Figures 1, 3 and 4: Kindly present (a), (b), (c) etc. in uppercase letters.

Discussion: The discussion of results is poor. Kindly improve it. Authors are also advised to compare the results with literature.

Tables 1-5: Standardize the number of decimal places.

Concussion: Kindly include the future prospects of the present study.

The English and grammar need to be polished.

Reviewer 2 Report

The authors studied for devleopment of UPLC-MS/MS for quantitation of Jin-Gu-Lian (JGL) and its core drug pair, and then characterized their pharrmacokinetics. JGL contains Salidroside, Anabasine, Chlorogenic acid, and Protocatechuic acid, the authors determined the four comounds in rat plasma for pharmacokinetic analysis. However, the developed LC-MS/MS method for the compounds has not novelty, and their findings could be not suite for publication in this journal. I recommend that the manuscript should be submitted to the journals related to bioassay and its method develompment or validation. Additionally, the manuscript has some points to revise.

Major points,

The author should discuss the previous reported literature about the development of LC-MS/MS bioassay of Sal, Ana, Ca, or PCA, and state what difference or novelty is.

Minor points,

The author shoud give the information with same order for reader. In the manuscript gave the compound order with Sal, Ana, CA, PCA, but in some figure Ana, PCA, Sal, CA, it would be confused.

The LLOQs are 13.10, 0.1125, 1.85, and 1.6172 for Sal, Ana, CA, and PCA, respectively. Please give the reason to give the values. In general, the LLOQs usually use simple number like 0.1, 1, 10… ng/mL. Any special reason ? The upper ranges are also specific value.

I would be better that In Table 4, please show the conditions along with temperature and incubation time like discribe in the manuscript.

Reviewer 3 Report

The current study established and validated a simple, rapid, and sensitive UPLC-MS/MS method for the simultaneous determination of four bioactive components in rat plasma. The approach was successfully applied to study the plasma pharmacokinetics following oral administration of Jin-Gu-Lian and its core drug pair extract, respectively.  The research topics seems interesting, however, the following points need to be addressed:

The manuscript needs to be revised by a native English speaker for several typo errors and grammatical mistakes.

Line 134: correct "coecients" to "coefficients"

Figure 3: the font of the peaks' labels need to be increased.

The nature of  Jin-Gu-Lian formula is not clear. the author described the composition of JGL formula "Sargentodoxa cuneata (Oliv.) Rehd. et W (SC), Alangium chinense(Lour.)Harms (AC), Gaultheria yunnanensis(Franch.)Rehd (GY), Schefflera leacantha Vig. (SL), and Psammosilene  tunicoides W.C.Wu & C.Y.Wu (PT)" without referring whether they were dried plant material or dried extracts and the proportion of each component.

Why did the authors prepared the core drug pair extract by boiling the dried SC and AC in water rather than extraction by percolation in methanol? This needs to be justified.

Though in their previous studies four compounds [salidroside, chlorogenic acid, liriodendrin and quercetin] were identified as chemical markers for quality control of Jin-Gu-Lian capsules. However, liriodendrin and quercetin were replaced by Anabasine (Ana), and Protocatechuic Acid (PCA), in the current study. This needs to be justified.

"Zhou ZY, Huang Y, Xiao JC, Liu H, Wang YL, Gong ZP, Li YT, Wang AM, Li YJ, Zheng L. Chemical profiling and quantification of multiple components in Jin-Gu-Lian capsule using a multivariate data processing approach based on UHPLC-Orbitrap Exploris 240 MS and UHPLC-MS/MS. J Sep Sci. 2022 Mar;45(6):1282-1291. doi: 10.1002/jssc.202100762. Epub 2022 Feb 1. PMID: 35060338." 

Round 2

Reviewer 1 Report

The authors have addressed the previous concerns. The revised manuscript is suitable for publication in its present form.

Reviewer 2 Report

The authors provided responses well for the concerns and corrected the previous manuscript well.